# Epidemiology and Emerging Trends of Zoonotic Viral Diseases of Pigs in India

**DOI:** 10.3390/v17030381

**Published:** 2025-03-06

**Authors:** Swaraj Rajkhowa, Joyshikh Sonowal, Seema Rani Pegu, Rajib Deb, Vivek Kumar Gupta

**Affiliations:** 1ICAR-National Research Centre on Pig, Rani, Guwahati 781131, India; drseemapegu@yahoo.com (S.R.P.); drrajibdeb@gmail.com (R.D.); gupta.drvivek@gmail.com (V.K.G.); 2Krishi Vigyan Kendra, Assam Agricultural University, Sribhumi 788712, India

**Keywords:** zoonotic viruses, pigs, Japanese encephalitis virus, Nipah virus, swine influenza, emerging zoonoses, India

## Abstract

Pigs serve as critical reservoirs and amplifiers for numerous zoonotic viral diseases, presenting substantial public health challenges in India. This study highlights the epidemiology and emerging trends of key zoonotic viruses associated with pigs, emphasizing their role in endemic and emerging disease dynamics. Japanese encephalitis virus (JEV) persists as a major concern, with pigs acting as amplifying host, while hepatitis E virus (HEV) remains a prominent cause of viral hepatitis, transmitted via contaminated water and pork products. Emerging high-fatality viral zoonoses caused by Nipah virus (NiV) and recurrent threats from swine influenza virus (SIV) demonstrate that the zoonotic landscape is evolving. Furthermore, zoonotic viruses like rotavirus, pseudorabies (ADV or SuHV-1), porcine astrovirus (PAstV), and Torque teno sus virus (TTSuV) reflect the expanding diversity of pig-associated pathogens in India. Emerging evidence also implicates viruses such as Chandipura virus (CHPV) in localized outbreaks, indicating broader zoonotic potential. Novel risks such as swine acute diarrhea syndrome coronavirus (SADS-CoV) and SARS-CoV-2 emphasize the role of pigs as potential intermediaries for pandemic-prone viruses. This comprehensive study evaluates the prevalence, outbreak dynamics, and public health implications of zoonotic viral diseases of pigs in India, providing valuable direction for developing effective control measures.

## 1. Introduction

Pigs occupy a unique position in global agriculture and public health due to their economic importance and role as reservoirs for numerous zoonotic pathogens. The intensification of pig farming and increasing human–pig interactions have heightened the risk of zoonotic disease transmission, thereby necessitating a comprehensive understanding of the associated health risks, as well as their emerging trends. The dynamic interplay of environmental, agricultural, and societal factors has contributed to the emergence and re-emergence of zoonotic diseases linked to pigs [1]. Pigs act as amplifying hosts for a wide array of pathogens, with zoonotic viral diseases posing the most significant challenges due to their ability to cause large-scale outbreaks and cross-species transmission [2]. Climate change, land-use alterations, and expanding vector habitats further exacerbate these risks, increasing the likelihood of novel disease outbreaks and the re-emergence of pathogens [3]. The growing evidence of potential zoonotic viruses in neighboring regions stresses the importance of inclusion of surveillance systems and a proactive One Health approach to sense and spot the threat of zoonoses in India. Table 1 highlights the prevalent zoonotic viruses of pigs in India, emphasizing the urgent need for targeted research and intervention strategies to address the risks associated with these pathogens.

## 2. Major Zoonotic Viral Diseases of Pigs in India

### 2.1. Japanese Encephalitis Virus (JEV)

Japanese encephalitis virus (JEV) is a zoonotic flavivirus that poses a significant public health challenge in India [4]. First identified in the Vellore district of Tamil Nadu in 1955, the virus has since spread across various states, becoming a perennial threat [5]. JEV is primarily transmitted by *Culex species* mosquitoes, which thrive in waterlogged regions and rice paddies, conditions prevalent in rural India [6]. Pigs and water birds act as amplifying reservoirs, facilitating the virus’s enzootic cycle, while humans serve as dead-end hosts [7]. The virus manifests in humans as acute encephalitis, often leading to severe neurological symptoms, including seizures, disorientation, and coma [8]. Regular outbreaks in states like Assam, Manipur, Uttar Pradesh, Bihar, Haryana, Maharashtra, Goa, Karnataka, Kerala, Tamil Nadu, Pondicherry, Orissa, and West Bengal underscore its public health significance, particularly in regions with high pig populations and conducive mosquito-breeding environments [9,10].

JEV infections in pigs often go unnoticed due to the asymptomatic nature of the disease, though reproductive issues like abortion and stillbirths can occur in some cases [11]. Pigs play a pivotal role in amplifying the virus, heightening zoonotic risks, and potentially causing economic losses in the swine industry [12]. Diagnostics for JEV in pigs rely on a combination of clinical signs, serological tests like enzyme-linked immunosorbent assay (ELISA), and molecular techniques such as RT-PCR for accurate detection [13]. Notably, JEV can be classified into five genotypes (Genotype I to Genotype V) based on the diversity of the nucleotide sequence of the Envelope (E) protein gene, with most isolates in India falling under Genotype I (GI) or Genotype III (GIII) [14]. This genetic diversity has significant implications for vaccine design and control strategies. Recent advancements in research, including genomic studies and surveillance systems, aim to improve disease control. ICAR-IVRI, Izatnagar, has developed a Vero cell culture-based inactivated JEV vaccine for pigs and filed a patent, marking a significant advancement in JEV control [15]. Despite the introduction of the SA-14-14-2 and JENVAC vaccines for humans, gaps remain in the prevention of JEV transmission from amplifying hosts like pigs to humans. The rising incidence of Japanese encephalitis in India highlights an urgent need for targeted interventions and improved infrastructure for early detection and prevention of this important zoonotic disease in India.

### 2.2. Hepatitis E Virus (HEV)

HEV is a significant cause of acute viral hepatitis in India, with Genotype IV being commonly associated with pigs [16]. Transmission occurs through the fecal–oral route, consumption of undercooked pork, and/or contaminated water [17]. Enhanced public awareness and improved sanitation are critical for reducing HEV’s burden. HEV is a small, non-enveloped RNA virus belonging to the family *Hepeviridae*, genus *Orthohepevirus* [18,19]. It is a leading cause of acute viral hepatitis worldwide, with significant zoonotic potential. Among animals, pigs are considered the primary reservoir for zoonotic HEV, particularly genotypes HEV-3 and HEV-4, which can infect both animals and humans [20]. Transmission typically occurs via the fecal–oral route, and HEV is detectable in the feces, bile, and muscles of infected pigs [21]. Although the risk of contracting HEV from pigs is generally low, zoonotic transmission can occur through the consumption of raw or undercooked pork. Other risk factors include direct contact with infected animals or their feces, and susceptibility increases in immunocompromised individuals and pregnant women [22].

Studies conducted in India highlights the prevalence and risk factors associated with HEV in swine populations. Milton et al. (2023) reported a 51% seroprevalence in backyard pigs in northeastern India, identifying the risk factors such as swill feeding, lack of disinfection, and mixed-age rearing practices [23]. Similarly, Bansal et al. (2017) recorded a 65% seroprevalence of JEV in the state of Punjab, with a higher prevalence in growing pigs (2–8 months old), suggesting age-related susceptibility [24]. Swill-fed pigs also exhibited a higher prevalence of HEV RNA, underlining the importance of feed hygiene in reducing infection rates. Arankalle et al. (2001) reported seropositivity rates ranging from 54.6% to 74.4% in pigs across various Indian states [25], while Shukla et al. (2006) identified genetic distinctions between human and swine HEV isolates, suggesting limited zoonotic transmission in endemic regions [26]. The genetic diversity of swine HEV isolates underscores the need for targeted surveillance and improved biosecurity measures to mitigate risks, especially for occupationally exposed groups such as swine farmers, slaughterhouse workers, and veterinarians.

### 2.3. Swine Influenza Virus (SIV)

Swine influenza A virus (SIV) is a zoonotic pathogen of significant public health importance, causing respiratory disease in pigs and posing a potential risk of cross-species transmission to humans [27]. SIV is classified into subtypes based on its surface glycoproteins, hemagglutinin (HA), and neuraminidase (NA), with H1N1, H1N2, and H3N2 being the most commonly reported subtypes in swine populations globally [28]. In India, limited but significant studies have reported the presence of SIV in pigs, primarily associated with sporadic outbreaks of respiratory illness among swine herds [29,30]. Surveillance reports indicate varying prevalence rates depending on the region and diagnostic methods employed. Nagarajan et al. (2010) reported the detection of a novel reassortant influenza A H1N1 virus in pigs in Uttar Pradesh, India, linked to the 2009 human pandemic strain [29]. Virus isolation, RT-PCR, and genetic analysis revealed that the HA gene originated from the “North American Swine” lineage, while the NA and M genes were from the “Eurasian Swine” lineage. Other genes were also of “North American Swine” origin. This was the first report of such a reassortant in Indian pigs, emphasizing the need for ongoing surveillance. Interestingly, the virus also demonstrated the capability for human-to-human transmission without requiring pigs as intermediate hosts [31]. Senthilkumar et al. (2021) conducted a serological survey on H1N1 infection in Indian pigs between 2009 and 2016, revealing seroprevalence rates ranging from 5.2% in 2009 to 36.3% in 2011 [30]. In eastern Uttar Pradesh, antibody prevalence varied from 6.2% to 37.5%, suggesting co-circulation of seasonal and pandemic H1N1 viruses. These findings underscore the potential for spillover transmission of influenza from humans to pigs, highlighting the need for systematic molecular surveillance to monitor and mitigate pandemic risks. Pegu et al. (2017) conducted SIV seroepidemiological studies in northeastern states, reporting seropositivity rates of 7.43% in Assam (49/659), 6.25% in Nagaland (7/112), 6.94% in Arunachal Pradesh (5/72), 8.16% in Meghalaya (8/98), 7.46% in Mizoram (5/67), and 5.36% in Tripura (3/56), with an overall positivity rate of 3.94% (42/1064 samples) [32]. These studies highlight the importance of monitoring and controlling SIV in the region. Infected pigs may exhibit symptoms such as coughing, sneezing, and respiratory distress. The prevalence of SIV in Indian pigs underscores the necessity for enhanced surveillance and biosecurity measures. The zoonotic potential of SIV, coupled with its ability to reassort and generate novel strains, highlights the importance of monitoring swine populations for emerging variants.

### 2.4. Rotavirus (RV)

Rotavirus, a non-enveloped double-stranded RNA virus belonging to the family *Reoviridae*, is a leading cause of enteric diseases in neonatal and young animals, including pigs [33]. In pigs, rotavirus primarily causes diarrhea and gastroenteritis, resulting in significant economic losses due to morbidity, mortality, and reduced growth rates [34]. Rotavirus species A, B, and C (RVA, RVB, RVC) are known to infect humans and animals, while species D, E, F, and G (RVD, RVE, RVF, RVG) primarily infect animals, mostly birds [35,36]. Epidemiologically, RVA is the most critical cause of diarrhea, with 27 G types and 35 P types identified so far [36]. RVA strains are categorized based on antigenic properties, RNA migration patterns, and nucleotide sequences [37].

Rotavirus-associated diarrhea in piglets is a significant concern in India, with varying prevalence and notable genetic diversity observed in different regions. Malik et al. (2013) detected RV in 13.04% of diarrheic piglets from various regions of Madhya Pradesh, using ELISA, RNA-PAGE, and RT-PCR as diagnostic tools [38]. Studies highlight the widespread nature of rotavirus infections, particularly in northeastern states such as Assam and Tripura, where prevalence rates reached up to 40.62% [39]. Barman et al. (1998, 2003) reported high positivity rates using ELISA, with 26.9% of samples testing positive for RV and the highest incidence observed in 4-week-old piglets (53.1%) [40,41]. Bora et al. (2007) isolated rotavirus from diarrheic piglets, showing that ELISA detected RV antigen in slightly higher numbers (29.35%) compared to PAGE (24.84%) [42]. Dubal et al. (2013) identified an infrequent strain, G4P [6], and reported 30% prevalence in the Northeastern Hills region [43]. Kattoor et al. (2018) reported emerging VP6 genotypes (I1, I5) and a positivity rate of 42.4%, with the highest detection in Uttar Pradesh [44]. Kusumakar et al. (2010) described significant genomic diversity with eight distinct electropherotypes, predominantly of the long pattern [45]. Sharma et al. (2013) detected a 46.3% prevalence of group A rotavirus, confirmed through electropherotype patterns [46]. Ahmed et al. (2017) highlighted zoonotic potential, identifying shared genotypes such as G3P [6] in piglets and humans in close contact [47]. Chakraborty et al. (2015) developed an RFLP assay to analyze 67 rotavirus isolates (29 Indian, 38 global) from humans, bovines, and porcine neonates. Using digestion of RT-PCR-amplified VP7 cDNAs with VspI, HaeIII, and NlaIV, they identified 48 RFLP patterns, 20 unique to Indian isolates, revealing significant genetic variation [48]. This emphasizes the critical need to understand the genetic diversity and interspecies transmission of rotavirus in India.

## 3. Emerging and Sporadically Reported Zoonotic Viruses of Pigs in India

### 3.1. Chandipura Virus (CHPV)

Chandipura virus (CHPV) is an enveloped, single-stranded RNA virus belonging to the family *Rhabdoviridae* and genus *Vesiculovirus* [49,50]. First isolated in India in 1965, CHPV has emerged as a significant neurotropic pathogen causing acute encephalitis, primarily in children [49]. Fruit bats and sandflies are well-established as reservoirs and vectors, respectively, while evidence regarding the involvement of pigs in CHPV’s epidemiology remains limited. Pigs are suspected amplifying hosts for vesiculoviruses, but their role in CHPV transmission remains underexplored, with limited epidemiological studies in India. Joshi et al. (2005) reported virus-neutralizing antibodies in 30.6% of pig sera from the Karimnagar and Warangal districts of Andhra Pradesh, indicating exposure to CHPV [51]. However, retrospective studies found N-antibodies to CHPV in pigs, indicating prior exposure, although no signs of illness were observed in the animals [52]. These findings suggest that pigs may not play a significant role as reservoirs or amplifying hosts for CHPV in India. Still, the possibility of spillover events, particularly in high-density farming areas or during outbreaks involving vectors such as sandflies, cannot be ruled out. Further studies, including serological surveys and vector–host interaction research, are essential to elucidate the role of pigs in CHPV transmission dynamics. Enhanced surveillance and interdisciplinary approaches are critical for understanding and mitigating the zoonotic risks posed by CHPV to animal and human health.

### 3.2. Rabies Virus

Rabies virus (RABV), a neurotropic virus belonging to the family *Rhabdoviridae* and genus *Lyssavirus*, is a significant zoonotic pathogen responsible for approximately 59,000 human deaths annually worldwide, with India contributing a substantial share of this burden [53]. While domestic dogs serve as the primary reservoirs and transmitters of rabies, the virus’s ability to infect other mammals, including livestock, raises concern about its occurrence in unconventional hosts such as pigs [1]. Although pigs are not recognized as primary reservoirs, they can act as incidental hosts when exposed to infected animals, particularly rabid dogs.

Reports of rabies in pigs are rare in India, reflecting its sporadic and opportunistic occurrence. Isolated cases have been documented in rural areas where pigs often roam freely and share habitats with rabies-endemic carnivores. For instance, Preethi et al. (2020) reported the first documented case of rabies in domestic pigs under farm conditions in South India, where three pigs from a pig farm in Kerala exhibited nervous symptoms 10 days after a stray dog attack, dying within 48 h, with rabies confirmed in one pig through immunochromatographic assay and fluorescent antibody testing (FAT) [54]. Similarly, Boro et al. (2023) reported rabies cases in pigs from northeast India, emphasizing the sporadic yet notable incidence of the disease in this species [55]. Kinge and Supe (2016) conducted a study at a tertiary care hospital in Nagpur, documenting rabies cases in humans following animal bites, with 0.6% of these cases attributed to pig bites [56]. Furthermore, Nair and Jayson (2016) reported a human–wild pig conflict in Malappuram District, Kerala, with rabies confirmed in affected areas and among victims [57]. These cases underscore the potential for rabies spillover to pigs in regions with high canine rabies prevalence. Although such occurrences are infrequent, they highlight the importance of including pigs in rabies surveillance programs. Preventive measures, such as vaccination campaigns targeting domestic dogs and educational initiatives for farmers, are crucial to minimize livestock exposure to rabid animals and control the risk of rabies transmission in unconventional hosts such as pigs.

### 3.3. Pseudorabies or ADV (SuHV-1)

Pseudorabies virus (PRV), also known as Aujeszky’s disease virus (ADV), is an enveloped, double-stranded DNA virus belonging to the family *Herpesviridae* and subfamily *Alphaherpesvirinae* [58]. It is a highly contagious pathogen primarily affecting pigs, its natural host, and causing respiratory, neurological, and reproductive disorders [59]. PRV also has a broad host range, with secondary hosts such as cattle, dogs, and cats often succumbing to fatal infections [60]. While pseudorabies has been eradicated in several developed countries through rigorous vaccination and control measures, its presence in India remains a concern due to sporadic reports of infection in swine populations. Studies on PRV prevalence in Indian pigs have underscored the necessity for enhanced surveillance and diagnostic measures. Molecular epidemiological investigations conducted by Vanamayya et al. (2009) aimed to optimize a SYBR Green-based real-time PCR assay for PRV detection in the Indian pig population [61]. A total of 1050 DNA samples from tissues of native Indian pigs and 80 nasal swabs from animals exhibiting respiratory illness were screened using both conventional PCR and an in-house validated SYBR Green real-time PCR assay. Notably, none of the samples tested positive for PRV. The status of pseudorabies prevalence in India remains unclear, with limited data available. Apart from the study by Bhattacharya et al. (1974), which involved virus isolation from swine, there is a lack of comprehensive epidemiological studies [62]. Previous research by Vanamayya (2002) recorded a seropositivity rate of approximately 21% in native pigs reared under a backyard system of management, while no seropositivity was detected in exotic breeds of farm-reared swine [61,63]. Although large-scale outbreaks have not been reported in India, the latent nature of PRV and its potential for reactivation under stress conditions pose a significant threat to swine health and productivity in India.

### 3.4. Porcine Astrovirus (PAstV)

Porcine astrovirus (PAstV) is a non-enveloped, single-stranded RNA virus belonging to the family *Astroviridae* [64]. It is a common enteric virus in pigs, typically associated with subclinical infections or mild diarrhea. PAstV is genetically diverse, with multiple species (PAstV1–PAstV5) identified globally [65]. Although often considered a low-impact pathogen, emerging evidence suggests its potential association with neurological disorders in pigs, raising concerns about its broader implications for swine health. In India, research on PAstV prevalence in pigs is limited but gradually gaining attention. Recent studies employing molecular diagnostics such as RT-PCR have detected PAstV in fecal samples from swine populations in regions with intensive pig farming. The prevalence and genetic diversity of porcine astrovirus (PAstV) in India have been highlighted in several recent studies, revealing significant heterogeneity and regional variability. Sawant et al. (2023) detected PAstV in 12.5% of fecal samples from piglets in Western Maharashtra, identifying strains belonging to MAstV-24, MAstV-26, and MAstV-31 [66]. This aligns with earlier findings by Kattoor et al. (2019), who first reported PAstV prevalence in India, detecting the virus in 17.6% of diarrheic piglets and highlighting co-infections with porcine rotavirus A [67]. Their phylogenetic analysis revealed lineages 2 and 4, with lineage 4 exhibiting significant genetic variability. Subsequent studies, such as those by Kour et al. (2021) and Kaur et al. (2023), have corroborated the circulation of diverse PAstV genotypes, particularly PAstV1, PAstV2, PAstV4, and PAstV5, in Haryana, with prevalence rates ranging from 16.47% to 50% across various age groups [68,69]. Vaishali et al. (2023) further emphasized the high detection rate of PAstV (50%) in pigs, particularly in those aged 3–6 weeks, and identified the circulation of lineages 2 and 4 [70]. Collectively, these studies underscore the endemic presence of PAstV across different regions of India, with a notable genetic diversity that necessitates further investigation into its epidemiology, clinical significance, and potential impact on porcine health management. Porcine astrovirus (PoAstV) is a pathogen with potential zoonotic implications; its genetic diversity and potential for recombination necessitate further investigation to assess interspecies transmission risks [71].

### 3.5. Torque Teno Sus Virus (TTSuV)

Torque teno sus virus (TTSuV) is a small, non-enveloped, single-stranded circular DNA virus classified within the *Anelloviridae* family [72]. It is widely recognized as a ubiquitous virus in swine populations and is categorized into two main species, TTSuV1 and TTSuV2 [73]. Although TTSuV itself is not directly associated with clinical disease, it is often implicated as a co-factor in porcine circovirus-associated diseases (PCVADs), where its high viral load may exacerbate clinical outcomes. The virus is transmitted through both vertical and horizontal routes, and its widespread prevalence in pigs has made it a significant subject of study in veterinary virology. In India, reports on TTSuV prevalence in pigs are limited but provide valuable insights. Subramanyam et al. (2019) reported the first detection of Torque teno sus viruses (TTSuVs) in India, identifying both *Iotatorquevirus* and *Kappatorquevirus* genera in 110 of 416 pig samples collected between 2014 and 2018 [74]. Using PCR and DNA sequencing, the study highlighted the circulation of these viruses across 12 Indian states, raising concerns about their potential to cross the species barrier and their relevance, particularly in the northeastern states of India, where pork consumption is high.

## 4. Other Emerging Viruses Not Reported in Pigs in India but Circulating in Neighboring Countries

Viruses such as swine acute diarrhea syndrome coronavirus (SADS-CoV), SARS-CoV-2, and Nipah virus (NiV) underscore the increasing need to monitor pigs as reservoirs for pandemic-prone pathogens.

### 4.1. Swine Acute Diarrhea Syndrome Coronavirus (SADS-CoV) and SARS-CoV-2

Swine acute diarrhea syndrome coronavirus (SADS-CoV) and severe acute respiratory syndrome coronavirus 2 (SARS-CoV-2) are both members of the *Coronaviridae* family, with zoonotic origins and significant implications for animal and human health [75]. Swine coronaviruses are categorized into different genera, with *Alphacoronavirus* including transmissible gastroenteritis virus (TGEV), porcine respiratory coronavirus (PRCV), porcine epidemic diarrhea virus (PEDV), and swine acute diarrhea syndrome coronavirus (SADS-CoV); *Betacoronavirus* represented by porcine hemagglutinating encephalomyelitis virus (PHEV); and *Deltacoronavirus* comprising porcine deltacoronavirus (PDCoV) [76]. PHEV, one of the earliest swine coronaviruses to be identified, remains prevalent and generally causes asymptomatic infections in pigs worldwide. TGEV, PEDV, and PDCoV are enteropathogenic viruses that result in acute gastroenteritis with similar clinical presentations in pigs of all ages, while recently emerging coronaviruses, such as PDCoV and SADS-CoV, were reported in 2014 in the United States and 2017 in China, respectively [76]. This virus causes severe diarrhea and high mortality in neonatal pigs, posing a threat to global swine industries. On the other hand, SARS-CoV-2, a beta-coronavirus responsible for the COVID-19 pandemic, has demonstrated a broad host range, raising concerns about reverse zoonosis in animal populations, including pigs.

In the Indian context, to date, there have been no confirmed reports of SADS-CoV in Indian pigs, indicating its absence or lack of prevalence in the country. However, given India’s extensive pig farming and proximity to bat habitats, there is a need for proactive surveillance to detect potential spillovers. Regarding SARS-CoV-2, studies during the COVID-19 pandemic investigated the possibility of reverse zoonosis in pigs. Testing of swine populations for SARS-CoV-2 RNA and antibodies has yielded negative results, suggesting that pigs are not significantly involved in the epidemiology of SARS-CoV-2 in India. Despite the absence of evidence for widespread infection, India’s dense livestock populations, coupled with human–animal interface dynamics, necessitate vigilance. Continuous monitoring for coronaviruses in pigs, especially in regions with high bat activity or human–pig interactions, is crucial for early detection and containment.

### 4.2. Nipah Virus (NiV)

Nipah virus (NiV) is an enveloped, single-stranded RNA virus classified within the family *Paramyxoviridae* and genus *Henipavirus* [77]. It is a highly pathogenic zoonotic virus responsible for severe respiratory and neurological diseases in humans and animals. Fruit bats of the *Pteropus* genus are recognized as the primary natural reservoirs of NiV [77]. Transmission to humans can occur through contact with infected animals, consumption of contaminated food, or direct person-to-person transmission [78].

In India, NiV outbreaks have predominantly been linked to human-to-human transmission and exposure to infected fruit bats rather than pigs. The first recorded outbreak occurred in 2001 in Siliguri, West Bengal, followed by a second outbreak in 2007 in Nadia district, West Bengal [79]. Both outbreaks were marked by high fatality rates and evidence of person-to-person transmission. More recent outbreaks in Kerala in 2018, 2019, and 2021 highlighted similar patterns [79]. The 2018 outbreak in Kozhikode district resulted in 17 fatalities out of 18 confirmed cases, with investigations tracing the source of infection to fruit bats contaminating date palm sap or fruits consumed by humans [80,81]. NiV is transmitted through multiple routes, including the consumption of raw date palm sap contaminated by bat saliva, urine, or feces; close contact with infected animals such as bats or pigs; and caregiving for infected individuals. Despite pigs being a documented amplifying host in outbreaks in Malaysia and Singapore, evidence from India does not support their role as a significant reservoir for NiV.

Recent studies have further reinforced the absence of Nipah virus among the pig population in India. A comprehensive study conducted in Mizoram tested over 1100 serum samples from healthy pigs, finding no anti-NiV IgG antibodies. Additionally, 1512 serum samples collected from pigs across various Indian states between 2009 and 2012 were negative for NiV in both Western blot and indirect ELISA assays [82]. These findings suggest that pigs are not a major reservoir for NiV in the country, and no clinical cases in pigs have been documented during these periods. The genetic diversity of NiV strains circulating in India, as identified through molecular studies, indicates similarities to strains in Bangladesh but with unique regional variations. This highlights the importance of genomic surveillance to monitor potential mutations and predict outbreak patterns.

## 5. Advancements in Molecular and Serological Tools for Zoonotic Virus Detection and Surveillance

Pigs serve as important reservoirs for several zoonotic viruses, necessitating precise and standardized detection and surveillance strategies to minimize or eliminate potential public health risks. To date, significant advancements have been made in molecular and serological diagnostic tools for zoonotic viral infections in swine, contributing to improved swine health monitoring and zoonotic disease control in India. Molecular techniques such as RT-PCR and real-time PCR enable rapid, sensitive, and specific virus identification. Serological assays, including ELISA, lateral flow assay (LFA), hemagglutination inhibition (HI), and virus neutralization tests (VNTs), are widely used for antibody detection and immune response evaluation. These advanced diagnostic tools have greatly enhanced disease surveillance, facilitated early outbreak detection, and strengthened control strategies. Moreover, molecular diagnostics allow for genetic characterization of viruses, aiding in epidemiological tracking and risk assessment.

Various molecular, serological, and culture-based tools have been developed and utilized for the detection and surveillance of zoonotic pig viruses in India. Among molecular tools developed in India, RT-PCR and real-time RT-PCR have been widely utilized for the detection of viruses in pigs, such as JEV, HEV, SIV, and RV [39,83,84,85,86]. RT-PCR is particularly effective for detecting JEV in blood, brain, and cerebrospinal fluid, while real-time RT-PCR enhances sensitivity and enables rapid detection [10]. Reverse transcription loop-mediated isothermal amplification (RT-LAMP) has also been developed for rapid detection of JEV and SIV, offering field-deployable capabilities [83,86,87]. Whole-genome sequencing and RNA envelope (E) gene RT-PCR are used for the molecular characterization of JEV, contributing to understanding virus evolution and transmission patterns [10]. Nested-multiplex RT-PCR facilitates the identification of rotavirus RNA, while VP7 and VP4 gene sequencing provide insights into the genetic diversity of rotaviruses [39].

Serological tools play a crucial role in epidemiological surveillance by detecting viral antibodies in pigs and assessing immunity levels. The plaque reduction neutralization test (PRNT) and virus neutralization test (VNT) serve as gold-standard assays for JEV, though they are labor-intensive and require live virus handling [83]. The hemagglutination inhibition (HI) test has been traditionally employed for JEV and SIV subtyping but has limitations due to cross-reactivity [86,88]. ELISA-based assays, including indirect IgG and IgM ELISAs, are extensively used for detecting JEV, HEV, and SIV antibodies [29,85,89,90]. A recombinant NS1 protein-based ELISA and lateral flow assay has demonstrated high sensitivity and specificity for JEV [89,90,91]. Additionally, agar gel immunodiffusion (AGID) is used for routine SIV surveillance, while lateral flow assays provide a rapid and cost-effective alternative for JEV detection at the point of care [86,91,92]. Indirect fluorescence assays (IFA) facilitate antibody-based surveillance and aid in research settings [88].

Cell culture and virus isolation methods remain essential for confirming viral infections and studying virus pathogenicity. JEV has been successfully isolated through inoculation into mice and Vero cells, allowing for phylogenetic analysis and genotype identification [93]. SIV is detected through virus isolation in embryonated chicken eggs and Madin–Darby canine kidney (MDCK) cell cultures, aiding in vaccine and antiviral development [86]. The hemagglutination (HA) test is employed to confirm the presence of SIV in clinical samples, supporting influenza surveillance programs [29]. These advanced molecular, serological, and culture-based diagnostic tools have significantly improved zoonotic virus surveillance in India, enhancing disease control measures. However, more research is needed to develop specific molecular and serological detection tools for other zoonotic pig viruses. Table 2 summarizes global strategies for managing and eradicating pig-associated zoonotic diseases, providing insights for disease control planning in India.

## 6. Financial Implications of Viral Zoonotic Diseases in Pig Farming

Viral zoonotic diseases in pigs pose substantial financial risks to smallholder farmers and the broader pork industry in India. Globally, zoonotic viruses such as influenza, Japanese encephalitis, Nipah virus, hepatitis E virus, and coronaviruses contribute to significant economic losses [94,95]. These diseases lead to direct financial impacts through livestock mortality, mandatory culling measures, and decreased productivity, further straining the livestock sector (Figure 1). Indirect impacts include market disruptions, trade restrictions, and decreased consumer demand, leading to price fluctuations and financial instability for farmers. Smallholder farmers, who form the backbone of India’s pig farming sector, are particularly vulnerable, as limited resources make it challenging to absorb these financial shocks. The costs associated with disease management—such as biosecurity measures, vaccination programs, and surveillance—place an additional strain on farmers. Inadequate compensation for culled animals further exacerbates their economic hardships. Moreover, public health concerns surrounding zoonotic outbreaks often amplify these financial burdens, as fear-driven shifts in consumption patterns negatively affect market stability. Restrictions on pork trade and movement further disrupt supply chains, reducing revenue generation across the industry Table 2.

## 7. Conclusions

The present review highlights the significant zoonotic risks posed by viral diseases in pigs in India, emphasizing the intricate interplay of epidemiology, genetic diversity, and public health concerns. Diseases such as Japanese encephalitis virus (JEV), hepatitis E virus (HEV), and swine influenza virus (SIV) underscore the critical role of pigs as reservoirs and amplifying hosts, necessitating robust surveillance systems and targeted interventions. The emergence of viruses like porcine astrovirus (PAstV) and the detection of Torque teno sus virus (TTSuV) further underline the need for comprehensive biosecurity measures to mitigate risks, especially for occupationally exposed groups. The sporadic detection of Chandipura virus (CHPV) and rabies in pigs highlights potential spillover risks in specific regions, while limited data on pseudorabies and other emerging viruses call for intensified research and monitoring. Although no direct evidence links pigs in India to significant outbreaks of Nipah virus (NiV), swine acute diarrhea syndrome coronavirus (SADS-CoV), or SARS-CoV-2, the proximity of pig farming to bat habitats necessitates proactive surveillance to prevent potential spillovers. Understanding the genetic diversity and zoonotic potential of pig-associated viruses is crucial for developing effective vaccines, diagnostic tools, and control strategies. To strengthen disease control, this review underscores the need for enhancing rural diagnostic capacity, improving farmer awareness programs, and establishing systematic surveillance networks in high-risk regions, particularly where pig farming overlaps with bat habitats, to mitigate emerging zoonotic threats effectively.

## Figures and Tables

**Figure 1 viruses-17-00381-f001:**
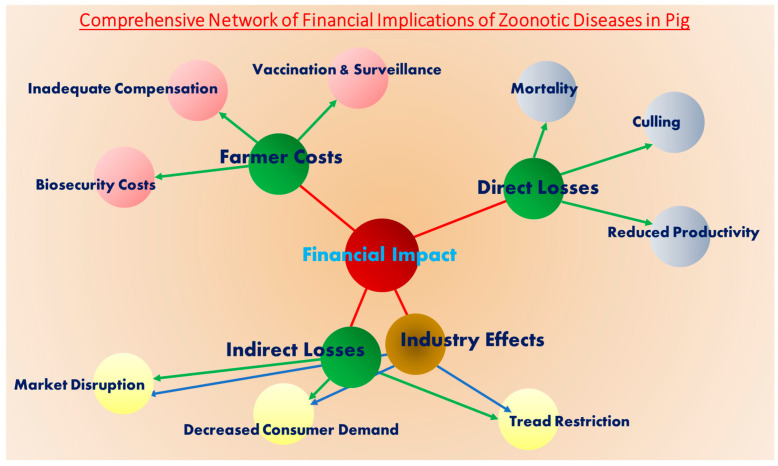
Diagrammatic Representation of the Financial Implications of Zoonotic Diseases in Pig Farming.

**Table 1 viruses-17-00381-t001:** Zoonotic viral diseases of pigs reported in India.

Pathogen	Etiology	Common Symptoms in Pigs	Route of Entry	Incubation Period	Transmission Cycle	Risk in Human
Japanese encephalitis virus (JEV)	Flavivirus	Fever, anorexia, neurological signs, reproductive failure	Mosquito bites (*Culex* spp.)	4–14 days	Enzootic cycle between mosquitoes and pigs, with humans as dead-end hosts	High in rural areas; severe neurological disease; zoonotic
Hepatitis E virus (HEV)	Orthohepevirus A	Subclinical infection, liver damage, reproductive issues	Fecal–oral route	2–8 weeks	Pigs act as reservoirs; humans infected via contaminated water or undercooked pork	High in regions with poor sanitation; causes hepatitis
Swine influenza virus (SIV)	Influenza A virus	Fever, cough, nasal discharge, lethargy	Inhalation	1–3 days	Direct pig-to-pig transmission; aerosolized respiratory secretions	Moderate; influenza-like illness, potential pandemic risks
Rotavirus (RV)	Rotavirus A, B, or C	Diarrhea, dehydration, weight loss	Fecal–oral route	1–3 days	Direct contact with feces; contamination of feed, water, or surfaces	Low; primarily affects children with gastroenteritis
Rabies virus	Lyssavirus	Neurological signs, aggression, paralysis	Bite wounds	2 weeks to 3 months	Animal bites or scratches; rare cases through aerosol transmission	High; fatal encephalitis in humans if untreated
Chandipura virus (CHPV)	Vesiculovirus	Fever, vesicular lesions, neurological signs	Bite from sandflies	3–7 days	Sandflies to pigs; potential spillover to humans	High; causes severe febrile illness and neurological symptoms
Pseudorabies/ADV (SuHV-1)	Suid alpha herpesvirus 1	Respiratory distress, neurological signs, reproductive failure	Inhalation or ingestion	3–7 days	Direct contact with infected secretions or contaminated feed	Low; humans are not natural hosts
Porcine astrovirus (PAstV)	Astrovirus	Mild diarrhea in piglets	Fecal–oral route	2–6 days	Direct fecal contamination of feed and water	Low; rare cases of human infection
Torque teno sus virus (TTSuV)	Anellovirus	Subclinical infection, potential immunosuppression	Fecal route and also found in feces, nasal excretions, sera, and liver of infected pigs	Unknown	Persistent infection; transmitted via bodily fluids	Low; no confirmed zoonotic cases
Nipah virus (NiV)	Henipavirus	Fever, respiratory distress, neurological signs	Ingestion, inhalation	4–14 days	Fruit bats to pigs; direct contact with infected pigs or contaminated materials; spillover to humans	High; severe encephalitis and respiratory illness

**Table 2 viruses-17-00381-t002:** Comparative global perspectives on zoonotic pig diseases.

Pathogen	India: Current Status and Challenges	Important Global Perspective (Examples of Successful Management)	References
Japanese encephalitis virus (JEV)	Endemic with seasonal outbreaks; pigs act as amplifying hosts; challenges include limited vaccination coverage and vector control.	Countries like Japan and South Korea have effectively controlled JE through widespread vaccination campaigns and robust mosquito control measures.	[96]
Hepatitis E virus (HEV)	Widespread, particularly in areas with inadequate sanitation; transmission linked to consumption of undercooked pork and contaminated water.	Improved sanitation, public awareness, and stringent food safety regulations have reduced HEV incidence in developed nations.	[97]
Swine influenza virus (SIV)	Regular occurrences, often underreported; lacks a structured surveillance system and vaccination strategy.	The USA and European countries manage SIV through continuous surveillance and routine vaccination of swine herds.	[98]
Rotavirus (RV)	High prevalence in piglets, leading to significant morbidity; vaccination programs are not widely implemented.	European countries and the USA have reduced piglet mortality through effective vaccination strategies and improved farm hygiene practices.	[99,100]
Rabies virus	Rare in pigs; sporadic cases reported, often due to bites from rabid animals; limited awareness and vaccination in pig populations.	Rabies control in livestock is achieved through vaccination of domestic animals and control of wildlife reservoirs in countries like the USA.	[101,102,103]
Chandipura virus (CHPV)	Occasional outbreaks with high fatality rates, primarily affecting children; pigs’ role in transmission is not well-established.	Limited global data; control measures focus on vector control and public health awareness in affected regions.	[104]
Pseudorabies (Aujeszky’s disease virus—SuHV-1)	Endemic in certain regions with sporadic outbreaks; control measures are limited and inconsistent.	Successfully eradicated in the USA and several European countries through comprehensive vaccination programs and stringent biosecurity protocols.	[59,105]
Porcine astrovirus (PAstV)	Presence in pig populations with unclear clinical significance; lack of routine surveillance and research.	Research in countries like the USA focuses on understanding pathogenicity and developing diagnostic tools.	[64,106]
Torque teno sus virus (TTSuV)	Commonly detected in pigs; clinical impact remains uncertain; no control measures in place.	In countries like the USA and parts of Europe, focuses on its co-infection dynamics, improved diagnostic methods, and potential immunization strategies, though no specific control measures are currently in place.	[107]
Nipah virus (NiV)	Occasional outbreaks with high mortality rates in human; surveillance in pig populations is inadequate.	Malaysia eliminated NiV from pig farms following the 1998–1999 outbreak by culling infected animals and implementing strict biosecurity measures.	[82]

## Data Availability

Data sharing is not applicable to this article, as no datasets were generated or analyzed during the current study.

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
