# Peer review of "Epidemiology and Emerging Trends of Zoonotic Viral Diseases of Pigs in India"

_viruses, 2025, doi:10.3390/v17030381_

Round 1

Reviewer 1 Report

Comments and Suggestions for Authors

The menuscript provides a comprehensive review of zoonotic viral diseases affecting pigs in India, focusing on both well-established and emerging pathogens. It is well-organized into targeted sections detailing individual viruses, their epidemiology, and their zoonotic implications. The inclusion of recent data and molecular epidemiological findings enhances the manuscript's relevance. Additionally, the focus on India's unique context and its juxtaposition with neighboring countries’ viral trends is a compelling aspect of the review. However, there are areas where the manuscript could be improved. 

1 . The manuscript provides a strong overview but could benefit from comparative insights connecting the epidemiology and management of pig-associated zoonotic diseases in India to global trends. For example, a deeper discussion of lessons learned from countries where some diseases (e.g., pseudorabies) have been eradicated would add value to policymakers in India.

2. The manuscript highlights molecular findings for many viruses (e.g., Rotavirus, TTSuV, and Porcine Astrovirus), it would be helpful to include a unified section or discussion synthesizing how advancements in molecular tools (e.g., RT-PCR, genetic sequencing) are influencing disease detection, surveillance, and mitigation strategies in pigs.

3. Although the conclusion discusses surveillance and biosecurity measures, the manuscript would benefit from practical policy suggestions such as: Enhancing diagnostic capacities in rural areas. Addressing gaps in farmer awareness or training. Establishing monitoring systems specifically targeting regions with mixed farming or bat habitats.

4. Given the economic importance of pigs in India, the manuscript could more thoroughly analyze the socio-economic impacts of zoonotic diseases. This would include financial losses, ramifications for smallholder farmers, and implications for India’s pork industry

5. The references appear comprehensive, but cross-checking is needed to ensure proper citation styles are followed consistently throughout the manuscript. For instance: Replace “[77, 78]” with consistent citation formatting as per Viruses journal guidelines. Linking older and newer studies coherently (e.g., Bhattacharya et al., 1974, and recent studies) can better demonstrate trends over time.

6. The addition of tables/figures summarizing data trends (e.g., prevalence rates by state) or showing phylogenetic trees for important viruses (e.g., Rotavirus, Nipah virus) would enhance readability.

7. Since the manuscript includes many technical terms (e.g., "electropherotypes," "genomic diversity," "Iotatorquevirus"), including a glossary or brief explanations (in footnotes or text) would make this accessible to a wider audience.

Comments on the Quality of English Language

1. In certain areas, there is a need for smoother transitions between sentences. Examples: Section 3.1 (Chandipura Virus): "Given the zoonotic potential of vesiculoviruses, pigs have been hypothesized...their role in CHPV transmission is underexplored in India." This could be made more concise. Section 3.2 (Rabies Virus): Simplifying verbose sentences will enhance flow. For instance, "Reports of rabies in pigs in India are rare, reflecting the sporadic and opportunistic nature of such infections," could be integrated better with subsequent details on cases.

2. Abbreviations - Full forms of abbreviations should be introduced upon first use for clarity (e.g., RT-PCR, RVA, RNA-PAGE).

Author Response

Dear Esteem Reviewer,

We sincerely appreciate your valuable time and constructive comments on our manuscript titled "Epidemiology and Emerging Trends of Zoonotic Viral Diseases of Pigs in India." Your insights have helped us refine the manuscript further. Below, we provide our responses to each comment along with the revisions made.

Comments and Suggestions for Authors

The manuscript provides a comprehensive review of zoonotic viral diseases affecting pigs in India, focusing on both well-established and emerging pathogens. It is well-organized into targeted sections detailing individual viruses, their epidemiology, and their zoonotic implications. The inclusion of recent data and molecular epidemiological findings enhances the manuscript's relevance. Additionally, the focus on India's unique context and its juxtaposition with neighboring countries’ viral trends is a compelling aspect of the review. However, there are areas where the manuscript could be improved. 

  1. The manuscript provides a strong overview but could benefit from comparative insights connecting the epidemiology and management of pig-associated zoonotic diseases in India to global trends. For example, a deeper discussion of lessons learned from countries where some diseases (e.g., pseudorabies) have been eradicated would add value to policymakers in India.

Response: We have now expanded the discussion to include comparative insights from countries that have successfully managed or eradicated pig-associated zoonotic diseases in a tabular form. This portion has been incorporated into the revised manuscript under the "Table 2: Comparative Global Perspectives on Zoonotic Pig Diseases" section.

  1. The manuscript highlights molecular findings for many viruses (e.g., Rotavirus, TTSuV, and Porcine Astrovirus), it would be helpful to include a unified section or discussion synthesizing how advancements in molecular tools (e.g., RT-PCR, genetic sequencing) are influencing disease detection, surveillance, and mitigation strategies in pigs.

Response: We have added a new section, "5. Advancements in Molecular and Serological Tools for Zoonotic Virus Detection and Surveillance" where we discuss molecular and other diagnostic innovations in India.

  1. Although the conclusion discusses surveillance and biosecurity measures, the manuscript would benefit from practical policy suggestions such as: Enhancing diagnostic capacities in rural areas. Addressing gaps in farmer awareness or training. Establishing monitoring systems specifically targeting regions with mixed farming or bat habitats.

Response: We have revised the conclusion to incorporate targeted policy recommendations, including: Strengthening diagnostic capacity in rural and smallholder pig farms, enhancing farmer education on zoonotic risks, Establishing systematic surveillance networks in high-risk regions (e.g., areas with close pig-bat-human interactions).

  1. Given the economic importance of pigs in India, the manuscript could more thoroughly analyze the socio-economic impacts of zoonotic diseases. This would include financial losses, ramifications for smallholder farmers, and implications for India’s pork industry

Response: We acknowledge the reviewer's valuable suggestion. However, data on the socio-economic impact of individual zoonotic pig diseases in India is limited. Nevertheless, we have incorporated a paragraph discussing the financial burden of these diseases, particularly their effects on smallholder farmers and the broader pork industry in India. This addition aims to provide a clearer understanding of the economic implications within the available scope of data.

  1. The references appear comprehensive, but cross-checking is needed to ensure proper citation styles are followed consistently throughout the manuscript. For instance: Replace “[77, 78]” with consistent citation formatting as per Viruses journal guidelines. Linking older and newer studies coherently (e.g., Bhattacharya et al., 1974, and recent studies) can better demonstrate trends over time.

Response: All references have been carefully cross-checked and reformatted to align with Viruses journal citation style.

  1. The addition of tables/figures summarizing data trends (e.g., prevalence rates by state) or showing phylogenetic trees for important viruses (e.g., Rotavirus, Nipah virus) would enhance readability.

Response: We have added two new tables summarizing data on major pig zoonotic viruses, and a figure has also been included.

  1. Since the manuscript includes many technical terms (e.g., "electropherotypes," "genomic diversity," "Iotatorquevirus"), including a glossary or brief explanations (in footnotes or text) would make this accessible to a wider audience.

Response: A glossary of key technical terms has been added as a supplementary file 1 to aid readability for a broader audience.

 Comments on the Quality of English Language

  1. In certain areas, there is a need for smoother transitions between sentences. Examples: Section 3.1 (Chandipura Virus): "Given the zoonotic potential of vesiculoviruses, pigs have been hypothesized...their role in CHPV transmission is underexplored in India." This could be made more concise. Section

3.2 (Rabies Virus): Simplifying verbose sentences will enhance flow. For instance, "Reports of rabies in pigs in India are rare, reflecting the sporadic and opportunistic nature of such infections," could be integrated better with subsequent details on cases.

Response: Transitions between sentences have been improved, particularly in sections discussing Chandipura Virus and Rabies Virus.

  1. Abbreviations - Full forms of abbreviations should be introduced upon first use for clarity (e.g., RT-PCR, RVA, RNA-PAGE).

Response: Abbreviations has been introduced in full upon first use as well as added as a supplementary file 1 to aid readability for a broader audience.

We appreciate the reviewer’s insightful suggestions, which have significantly enhanced the clarity, depth, and impact of our manuscript. We hope that the revised version meets the expectations for publication.

Reviewer 2 Report

Comments and Suggestions for Authors

Based on the manuscript  and in order to achieve the objective, It is necessary to provide additional epidemiological information that supports some of the statements in the document.

The zoonotic risk exists but a categorization or selection of those viral diseases of major relevance in India could be useful.

I suggest to provide additional information about  occurrence of these viral diseases in human and animals, clarifying the role of pigs in this interphase. It is also advisable to include an analysis of the risks and factors that represent a major threat. An approach from the one health perspective is necessary. 

The discussion section needs a considerable improvement.

Author Response

Response to Reviewer

Dear Esteemed Reviewer,

Thank you for your valuable feedback and insightful suggestions. We sincerely appreciate your time and effort in reviewing our manuscript. Below are our responses to your comments:

Reviewer Comment 1: 

Based on the manuscript and in order to achieve the objective, it is necessary to provide additional epidemiological information that supports some of the statements in the document. 

Response:  We have incorporated additional data, as reported by researchers, on Indian pigs affected, infected, or having survived zoonotic diseases. Furthermore, we have included details on the detection and surveillance tools developed within the country to enhance disease monitoring. 

Reviewer Comment 2: 

The zoonotic risk exists, but a categorization or selection of those viral diseases of major relevance in India could be useful. 

Response: 

We have categorized and included major zoonotic viruses relevant to India, particularly those with significant fatality rates and those detected in pigs. These additions can be found in Section 2 and Section 3 of the manuscript.

Reviewer Comment 3: 

I suggest providing additional information about the occurrence of these viral diseases in humans and animals, clarifying the role of pigs in this interphase. It is also advisable to include an analysis of the risks and factors that represent a major threat. An approach from the One Health perspective is necessary. 

Response:  In this manuscript, we have focused solely on zoonotic viral diseases reported in pigs from India and their disease dynamics. Additionally, we have analyzed key risk factors and major threats contributing to zoonotic spillover. A One Health perspective has been integrated as additional discussion and conclusion sections to emphasize the interconnectedness of human, animal, and environmental health.

Reviewer Comment 4: 

The discussion section needs considerable improvement. 

Response: The discussion has been significantly revised to provide a more comprehensive analysis.

Once again, we extend our sincere gratitude for your thoughtful review and constructive comments. Your feedback has greatly helped in strengthening our manuscript.
